# Soil Organic Matter Molecular Composition Shifts Driven by Forest Regrowth or Pasture after Slash-and-Burn of Amazon Forest

**DOI:** 10.3390/ijerph20043485

**Published:** 2023-02-16

**Authors:** Otávio dos Anjos Leal, Nicasio T. Jiménez-Morillo, José A. González-Pérez, Heike Knicker, Falberni de Souza Costa, Pedro N. Jiménez-Morillo, João Andrade de Carvalho Júnior, José Carlos dos Santos, Deborah Pinheiro Dick

**Affiliations:** 1Institute of Bio- and Geosciences—Agrosphere (IBG-3), Forschungszentrum Jülich GmbH, Wilhelm-Johnen-Straße, 52428 Jülich, Germany; 2Mediterranean Institute for Agriculture, Environment and Development-MED, Universidade de Évora, Ap 94, 7002-554 Évora, Portugal; 3Instituto de Recursos Naturales y Agrobiología de Sevilla (IRNAS-CSIC), Av. Reina Mercedes 10, 41012 Seville, Spain; 4Embrapa Acre, Rodovia BR-364, Km 14, Rio Branco 69900-970, Brazil; 5Departamento de Sistemas Físicos, Químicos y Biológicos, Universidad Pablo de Olavide, Ctra. Utrera, 1, 41013 Seville, Spain; 6Departamento de Energia, Universidade Estadual Paulista, Av. Ariberto Pereira da Cunha, 333, Portal das Colinas, Guaratinguetá 12516-410, Brazil; 7Laboratório Associado de Combustão e Propulsão, Instituto Nacional de Pesquisas Espaciais (INPE), Rodovia Presidente Dutra, km 40, Cachoeira Paulista 12630-00, Brazil; 8Departamento de Química, Universidade Federal do Rio Grande do Sul, Av. Bento Gonçalves, 9500, Porto Alegre 91501-970, Brazil

**Keywords:** experimental burning, Brazilian Amazon, Brachiaria, n-alkanes, analytical pyrolysis, van Krevelen 3D diagrams, Acrisol

## Abstract

Slash-and-burn of Amazon Forest (AF) for pasture establishment has increased the occurrence of AF wildfires. Recent studies emphasize soil organic matter (SOM) molecular composition as a principal driver of post-fire forest regrowth and restoration of AF anti-wildfire ambience. Nevertheless, SOM chemical shifts caused by AF fires and post-fire vegetation are rarely investigated at a molecular level. We employed pyrolysis–gas chromatography–mass spectrometry to reveal molecular changes in SOM (0–10, 40–50 cm depth) of a slash-burn-and-20-month-regrowth AF (BAF) and a 23-year Brachiaria pasture post-AF fire (BRA) site compared to native AF (NAF). In BAF (0–10 cm), increased abundance of unspecific aromatic compounds (UACs), polycyclic aromatic hydrocarbons (PAHs) and lipids (Lip) coupled with a depletion of polysaccharides (Pol) revealed strong lingering effects of fire on SOM. This occurs despite fresh litter deposition on soil, suggesting SOM minimal recovery and toxicity to microorganisms. Accumulation of recalcitrant compounds and slow decomposition of fresh forest material may explain the higher carbon content in BAF (0–5 cm). In BRA, SOM was dominated by Brachiaria contributions. At 40–50 cm, alkyl and hydroaromatic compounds accumulated in BRA, whereas UACs accumulated in BAF. UACs and PAH compounds were abundant in NAF, possibly air-transported from BAF.

## 1. Introduction

The humid microclimate underneath the forest canopy is considered the main protection mechanism of the Amazon Forest (AF) against fire. Additionally, the dense canopy of the AF reduces the intensity of solar radiation reaching the necromass accumulated on the forest floor, thereby reducing the flammability of the forest [1]. Therefore, wildfires in the AF are unusual under natural conditions. Nevertheless, the vulnerability of AF to wildfire has increased largely due to human-induced disturbance of these natural anti-wildfire mechanisms, particularly via slash-and-burn of forest and its replacement with food crops (mainly soybean) and pastures [2,3,4]. By replacing the original forest with plants containing a shallower root system, lower evapotranspiration and thinner canopy, the resulting environment is more flammable and vulnerable to wildfire. Around 41% of fires registered in the AF in 2021 did not occur in post-slash-and-burn areas (as usually expected and human-induced), but in primary AF standing vegetation [5]. This can be attributed to the formation of so-called flammable edges, where the ignition source (agricultural fire) is close to the standing vegetation, which in turn has become more susceptible to fire after forest slash-and-burn practiced in neighboring sites.

Deforestation across the Amazon Basin and the Brazilian AF has reached 17 and 20%, respectively [6], with approximately 60% of the deforested Brazilian AF having been converted to pastures [7,8]. Finer and Vila [5] predict patterns where human-induced fires in cleared AF lands in wet seasons will be followed by fires in adjacent standing AF in the dryer periods. This scenario will likely be aggravated by climate-change-induced higher air temperature, and the occurrence of longer, hotter and dryer fire seasons in the AF leading to increased fire intensity and mortality of trees post-fire [9]. According to Brando et al. [10], climate change may double the wildfire-affected area by 2050 in the southern Brazilian AF. Identification and isolation of slash-and-burn AF sites where forest instead of pastures may be implemented may be the key to the concomitant reestablishment of a humid ambience, the forest canopy and water cycles and to ultimately reduce the vulnerability of the AF to wildfire. Brando et al. [11] reported rapid canopy formation (70–80% closure) and full recovery of net CO_2_ exchange and evapotranspiration in less than seven years of AF regrowth after an experimental fire.

Burning of the AF not only converts standing vegetation carbon (C) stocks into CO_2,_ thereby increasing global warming, but it also disturbs soil organic matter (SOM) dynamics [12,13], possibly turning AF soils—known as large terrestrial C sinks—into large CO_2_-emitting sources [10,14]. In a previous study, we found that in one hectare of native AF in Acre State, Brazil, 68% (191 Mg ha^−1^) of total organic C (TOC) was stored as aboveground forest biomass, while 32% (90 Mg ha^−1^) was stored as soil organic C (SOC) (Acrisol, 0–2 m depth) [15]. Marques et al. [16] reported similar SOC stocks (0–2 m depth) for an Oxisol (98 Mg ha^−1^), a Spodosol (81 Mg ha^−1^) and an Ultisol (73 Mg ha^−1^) of Brazilian AF.

The SOC stocks in the AF are strongly dependent on the constant addition of labile organic C to the soil and litter formation, which is initially incorporated into soil as particulate labile material [17]. Due to the high average annual air temperature and precipitation and the coarse texture of AF soils, SOM protection mechanisms such as occlusion into aggregates and association with minerals are impaired, while the activity of microorganisms is stimulated, accelerating the turnover of SOC [18]. For instance, Marques et al. [16] reported that in AF soils (0–40 cm depth), labile SOC (i.e., free light fraction) represented up to 80% of TOC, whereas below 40 cm depth it accounted for <20%. This agrees with our previous work, where the proportion of labile SOC (sand + silt fraction)/TOC in a native AF site decreased from 37% (0–5 cm) to 32% (5–10 cm), 24% (20–30 cm) and 23% (40–50 cm) [15].

During a forest fire, the litter and the SOM of shallow soil layers are exposed to thermochemical alterations as a result of combustion. Furthermore, partially charred fragments of fire-affected standing vegetation may be deposited on the soil surface after fires. Usually, fire leads to a depletion of ligno-cellulosic and labile microbial-derived compounds and neoformation of C-condensed structures, conferring wider hydrophobicity and aromaticity indexes to SOM [19,20,21,22,23]. Together with a reduced soil microbial activity caused by fire [24], this partially explains the higher mean residence time of SOM observed in fire-affected soils compared to control sites [25].

Pasture implementation in fire-affected AF soils reduces the input of fresh biomass to soil compared to the original forest condition and has been shown to further deteriorate soil attributes and resilience, hindering possible regenerative measures of the AF [26]. Soil δ^13^C signature studies have revealed that SOC derived from native AF can still contribute to a large portion of TOC stocks in long-term pasture soils [12,17,27,28], emphasizing that: (i) the recovery of SOC stocks after deforestation/forest fire attributed to pasture may be often overestimated; and (ii) the use of analytical techniques is essential to improve our understanding of SOM dynamics in AF fire-affected vegetated soils.

In a previous work, we used cross-polarization magic-angle spinning solid-state ^13^C nuclear magnetic resonance (CPMAS ^13^C NMR) analysis to investigate the effects of AF fire and post-fire pasture on SOM composition [15]. Compared to the native AF soil (unburned), we observed slightly higher contribution of Aryl-C to SOM composition in two slash-and-burn-affected soils (one two years after AF regrowth, and one after 23 years of *Brachiaria brizantha* pasture) at both 0–5 and 5–10 cm soil depth. Our results suggested that the massive root system of Brachiaria and the input of fresh organic material to the soil due to fast forest regrowth may have contributed to an enrichment of O-Alkyl C groups, thereby masking stronger Aryl-C signals which are usually attributed to pyrogenic C in fire-affected soils [20]. However, this interpretation remained unclear. Furthermore, we observed lower Aryl-C to Alkyl-C ratios at 0–5 and 5–10 cm soil depth in the AF regrowth site (0.61 and 0.33, respectively) compared to the pasture site (0.77 and 0.67, respectively), suggesting a comparatively greater deposition and incorporation of fresh plant residues into soil by forest regrowth only two years after the fire [29]. Overall, these findings highlight: (i) fire increased SOM aromaticity; (ii) early regrowth of AF increased SOM lability; and (iii) the potential of forest regrowth to quickly re-build below- and aboveground C stocks, as similarly reported by Martin et al. [30].

Interestingly, Figueiredo et al. [31] observed that new sources of available nitrogen (N) from burning of AF biomass stimulate gross mineralization and, thus, the fast regrowth of AF after fires. Consequently, the regrowth of the AF post-fire can lead to a fast recovery of SOC and aboveground C stocks, especially if the soil has not yet been used for pasture [4,18,32]. Accordingly, Poorter et al. [18] estimated that tropical secondary forests re-growing in abandoned agricultural areas can take up C approximately 11-fold faster than old-growth forests. Nevertheless, studies examining SOM composition shifts at the molecular level caused by fire and post-fire vegetation to accelerate restoration of the AF are scarce.

Pyrolysis–gas chromatography–mass spectrometry (Py-GC/MS) has been widely applied as a powerful technique to investigate the effects of wild and anthropogenic fires on SOM molecular composition. Recent literature has proposed the use of van Krevelen diagrams to compile and facilitate the interpretation of the complex molecular assemblages released by Py-GC/MS [33]. In Deus et al. [34], Py-GC/MS was used to identify diagnostic compounds explaining increased SOM hydrophobicity after a forest fire. This was found to be attributed to an enrichment in aromatic and condensed structures with low oxygenation, and to lipid compounds. In Almendros et al. [35], Py-GC/MS-derived van Krevelen diagrams of humic acids were produced to discern molecular characteristics of SOM in agricultural soils associated with soil resilience. Of particular interest for our study, Py-GC/MS-derived van Krevelen diagrams have also been effectively used to disentangle shifts in SOM composition caused by fire from those caused by post-fire rehabilitation actions [33,36].

In our previous study, we observed distinct SOM stocks and chemical composition (^13^C NMR spectra) of fire-affected AF soils compared to the native AF soils [15]. Therefore, in the present study, Py-GC/MS is combined with three-dimensional (3D) van Krevelen diagrams as an effective approach to complement our ^13^C NMR findings, as suggested by De la Rosa et al. [37]. The aim of this study is to identify changes in SOM molecular composition caused by AF fire and post-fire forest regrowth or pasture. Additionally, we aim to disentangle more clearly the effects of fire and forest regrowth on SOM composition and aromaticity, which remained unclear in our preliminary study. This may improve our understanding of SOM and soil quality recovery in fire-affected AF soils.

We hypothesize that Py-GC/MS will corroborate our ^13^C NMR study, revealing strong remaining effects of fire on SOM molecular composition despite the elapse of 20 months of AF regrowth since the fire event. Therefore, we expect to detect depletion of labile compounds (e.g., polysaccharides) and enrichment of less-labile compounds (e.g., aromatics and polycyclic aromatic hydrocarbons) in the fire-affected soil under AF regrowth. Fire-effects on SOM composition in the pasture site may be less pronounced due to the long-term cultivation of pasture after fire.

## 2. Materials and Methods

### 2.1. Experimental Area and Controlled Forest Burning

The study site is located in the Northern Region of Brazil, in the Brazilian western-most state Acre, Cruzeiro do Sul municipality, Santa Luzia settlement, western part of the Brazilian Amazon region (Figure 1). The climate is defined as Af (tropical humid without a dry season) according to Köppen’s classification [38]. The annual average temperature is 24.9 °C, annual average rainfall is 2280 mm and relative air humidity is 86%. The soil is classified as Acrisol according to IUSS Working Group WRB [39], and as *Argissolo Vermelho distrófico plíntico* according to the Brazilian Soil Classification System [40], with 80.6% sand, 13.6% silt and 5.8% clay at 0–5 cm depth; 77.7% sand, 15.2% silt and 7.1% clay at 5–10 cm depth; and 64.6% sand, 19.6% silt and 15.8% clay at 40–50 cm depth [15]. This soil type is the second most representative of the Amazon territory (33%) and the most representative of Acre state (38%).

The experimental slash-and-burn of AF (Appendix A) was conducted in July 2010, with the authorization of the competent Brazilian governmental agencies (i.e., *Instituto do Meio Ambiente do Acre* and *Secretaria do Meio Ambiente do Estado do Acre*). Slash-and-burn was performed in 4 ha of primary ombrophilous forest with palm trees. The area was selected as representative for the burning experiment and isolated to prevent spreading of fire. The total fresh aboveground biomass within the selected field was estimated as 688 Mg ha^−1^, from which 85% consisted of plants with diameter at chest height > 10 cm and 15% of litter and plants with diameter at chest height < 10 cm. The total C amount contained in the aboveground vegetation was estimated as 191 Mg ha^−1^. Assuming a total biomass consumption of 40%, the release of C from vegetation to the atmosphere attributed to burning was estimated as 74.3 Mg ha^−1^. Details about forest inventory, biomass consumption by fire and CO_2_ emissions can be found in Carvalho Jr. et al. [41].

### 2.2. Soil and Litter Sampling and Characterization

Soil sampling was performed in September 2012. Soil samples were collected at the burned AF site (BAF), at a native AF site—unburned (NAF) and at a pasture field cultivated with *Brachiaria brizantha* (BRA) for 23 years after AF burning. Appendix A show NAF, BAF and BRA studied sites, respectively. At the time of soil sampling, BAF was under natural forest regrowth since burning (July 2010). The BRA site was occupied by native AF until 1989, when slash-and-burn of AF was performed. Brachiaria pasture was established right after. According to local farmers, this was the only known fire event in BRA area until soil sampling. The BRA is characterized by continuous extensive grazing with low stocking rates and no fertilization.

Soil samples were collected from soil trenches opened at NAF, BAF and BRA sites. The soil trenches were located at same topographic position and belong to the same soil class (Acrisol) as reported in Leal et al. [15]. Substantial amounts of soil were collected from 0–5, 5–10 and 40–50 cm soil depth. The soil samples were air-dried and sieved at 2 mm.

Soil pH was measured in distilled water (soil:water, 1:1, *v*:*v*), available P and K^+^ were extracted using Mehlich-1 solution (0.05 M HCl + 0.0125 M H_2_SO_4_) in soil:solution ratio of 1:10 (*v*:*v*), and exchangeable Ca^2+^, Mg^2+^ and Al^3+^ were extracted with 1 M KCl in soil:solution ratio of 1:20 (*v*:*v*). The content of Ca^2+^, Mg^2+^, Al^3+^, K^+^ and P was determined by inductively coupled plasma optical emission spectroscopy (ICP-OES, Perkin Elmer 7300). Potential acidity (Al + H) was determined using a buffered SMP solution, and effective cation exchange capacity (CECe) was calculated as the sum of K^+^, Ca^2+^, Mg^2+^, Al^3+^. These analyses were conducted according to Tedesco et al. [42] for characterization of the samples (Table 1). The TOC content of the soil samples (Table 1) was determined by dry combustion (PerkinElmer 2400).

Litter samples were collected next to the soil trenches opened at BAF, NAF and BRA site. The litter samples were oven-dried at 50 °C, milled in a mortar and analyzed for total C and N content by dry combustion (Thermo Fisher Scientific—Flash EA1112) to determine their C:N ratio: BAF = 22:1, NAF = 23:1, BRA = 92:1.

### 2.3. Pyrolysis–Gas Chromatography–Mass Spectrometry (Py-GC/MS) Analysis

Analytical Py-GC/MS was performed in a microfurnace pyrolyzer (Frontier Labs. Fukushima, Japan. Mod. 2020i) coupled to a gas chromatograph (Agilent, Sta. Clara, CA, USA. Mod. 6890N). Well-mixed and finely ground soil samples (2 mg) were used for the analysis. The samples were placed in deactivated stainless-steel capsules (Frontier Labs. EcoCups) and introduced in the preheated microfurnace for direct flash pyrolysis at 500 °C during 1 min. The gases evolved during pyrolysis were separated using a fused HP-5MS-ui capillary column (30 m × 250 μm × 0.25 μm i.d.). The carrier gas was He at a controlled flow rate of 0.9 mL min^−1^. The initial oven temperature was held at 50 °C for 1 min and then increased up to 100 °C at 30 °C min^−1^, from 100 to 300 °C at 10 °C min^−1^ and then stabilized at 300 °C for 10 min using a heating rate of 20 °C min^−1^. The compounds were detected by mass spectrometry (Agilent, Sta. Clara, CA, USA. Mod. 5973 MSD) in full scan mode and the mass spectra acquired at 70 eV ionizing energy.

The pyrolysis compounds were identified by analyzing the mass fragments and by comparison with published and stored mass data (NIST14 and Wiley7 libraries). The *n*-alkane series was characterized by monitoring diagnostic ions (*m/z* 57 and 85). The identified organic molecules released by Py-GC/MS were categorized into 7 families with known biogenic origin as in Jiménez-Morillo et al. [36]: unspecific aromatic compounds (UACs), polysaccharide-derived (Pol), peptide- and protein-derived (Pep), lignin-derived (Lig), lipids (Lip), N compounds (N-comp) and polycyclic aromatic hydrocarbons (PAHs). The relative abundance of the different families was calculated based on the identified compounds with >0.2% of total chromatogram area. The relative abundance of the pyrolysis chemical families was relativized according to the SOC content of the sample (g/kg OC, previously obtained in Leal et al. [15], and reported in Table 1 for convenience).

The structural information of the Py-GC/MS compounds was displayed as 3D van Krevelen plots according to Almendros et al. [43]. For that, compound-specific O/C and H/C atomic ratios were obtained from the empirical molecular formulas inferred from the mass spectra and plotted in the *x*, *y* plane. The chromatographic yields for individual compounds were calculated as total abundances and plotted in a third dimension (*z* axis).

The Py-GC/MS-released compounds and graphical figures were analyzed descriptively and interpreted comparatively (NAF vs. BAF vs. BRA) as commonly performed in the recent literature [22,43,44,45].

## 3. Results and Discussion

### 3.1. Molecular Composition of SOM Assessed by Py-GC/MS

The Py-GC/MS total ion chromatograms for NAF, BAF and BRA samples at the different soil depths are displayed in Appendix A, respectively, and the lists of released compounds are provided in Appendix A, respectively.

The SOM composition of the NAF sample collected at 0–5 cm depth was dominated by Lip and UAC groups, 4.1 and 3.21 g/kg OC, respectively, along with Pol compounds, 3.14 g/kg OC (Table 2, Figure 2). In NAF, such abundance of Lip and Pol compounds indicate the presence of relatively fresh SOM, i.e., organic matter partially degraded by microorganisms mixed with recently deposited biomass. This reflects a typical SOM composition of undisturbed forests [33,46].

In our previous work [15], we hypothesized that an expected stronger Aryl C signal in the ^13^C NMR spectrum of the BAF sample (0–5 cm depth), generally attributed to pyrogenic products of fire, could have been masked by an increased abundance of more labile chemical compounds. Here, we attribute this masking effect to the high abundance of Lip and Pep compounds in this sample (Table 2). In fact, the C:N ratio of BAF litter of 22:1 corroborates the assumed addition of fresh biomass to soil, which usually reduces Aryl C intensity in ^13^C NMR spectra [20]. Similar litter C:N ratio was observed in the NAF site (23:1), indicating deposition of fresh biomass on soil surface in both NAF and BAF.

A detailed analysis of the Lip compounds released by Py-GC/MS, revealed the existence of terpene-like compounds in NAF, such as isoledene, calamenene and corocalene (Appendix A), likely considered typical additions of the forest canopy to SOM [47,48]. These compounds were not detected in BAF and BRA, which shows a shift in the biomass type added to soil in NAF compared to BAF and BRA. Furthermore, the absence of the aforementioned Lip compounds in the SOM of BAF, indicate that despite the similar litter C:N ratio of NAF and BAF, the SOM of BAF has a lower contribution of mature canopy compounds, which have probably been extensively consumed by fire. The analysis of the alkyl compound series in soil samples at 0–5 cm depth soil samples corroborates these results by illustrating the highest abundance of *n*-alkane C25–C31 in NAF compared to BAF and BRA (Figure 3a), which is typical of arboreal and shrubby vegetation contribution to SOM composition [49].

The considerable contribution of UACs to SOM composition in NAF samples, which was observed at all investigated soil depths (Table 2), may be explained by an active defunctionalization of lignin-like compounds (methoxyphenols) induced by microbial activity [50]. Waggoner et al. [51] recently suggested that natural degradation of lignin may occur by exposure to ultraviolet radiation, resulting in the production of aromatic compounds. For instance, the contribution of Lig compounds to SOM composition at 0–5 cm depth was higher in NAF (0.65 g/kg OC) than in BAF (0.37 g/kg OC) and BRA (0.31 g/kg OC) (Table 2), as shown by the presence of non-altered lignin-containing plant biomass. These compounds can in turn be the precursor for aromatic-like compounds formation according to the process aforementioned. Despite the considerable contributions of UACs and Lig compounds to the SOM of NAF at 0–5 cm depth, it is most likely that the SOM is essentially composed of partially fresh material [33].

Interestingly, a relatively high proportion of pyrogenic-like compounds (i.e., PAHs, 1.16 g/kg OC) was observed in NAF at 0–5 cm depth (Table 2). This unexpected finding can possibly be assigned to the existence of fire-affected organic material due to unknown fire(s), which is highly recalcitrant, i.e., very stable over time [52,53], and/or, more likely, to the transportation of particulate burned material (soot) from adjacent fire-affected areas [46,54,55]. The proximity of NAF site to BAF site (Figure 1) may support this interpretation. These findings highlight the need for studies on pyrogenic compounds in undisturbed AF sites derived from neighboring fire-affected areas, which has not—or rarely—been acknowledged.

In BAF (0–5 cm depth), the relative abundance of PAHs (1.57 g/kg OC) and UACs (6.03 g/kg OC) was remarkably higher than that observed in NAF and BRA (Table 2). A conspicuous depletion of Pol- and Lig-like compounds is also observed in BAF (1.95 and 0.37 g/kg OC, respectively) compared to NAF (3.14 and 0.65 g/kg OC, respectively) at 0–5 cm depth (Table 2). Similar shifts in SOM molecular composition due to recent forest fire have been reported for different environments [56,57,58]. This is also corroborated by a prior characterization of our soil samples using solid-state ^13^C NMR [15]. Particularly, the increase in UACs (Table 2) together with the depletion of methoxyphenol (Lig) compounds observed in BAF (0–5 cm depth) (Appendix A), suggests that fire has triggered a defunctionalization of Lig compounds, as has been observed by other authors [23,46]. Alternatively, the origin for the UACs may also be the chemical alteration of terpene-like compounds and/or the cyclization of alkyl compounds through the Diels–Alder reaction [46]. For instance, we observed that these compounds disappeared in BAF (0–5 cm depth, Appendix A), possibly explaining, in part, the increased relative abundance of PAHs in BAF compared to NAF and BRA at 0–5 cm soil depth (Table 2).

The increased abundance of Lip compounds in BAF compared to NAF and BRA (0–5 and 5–10 cm depth) may be explained by the deposition of partially burned biomass on soil due to post-fire plant stress [19,59]. According to González-Pérez et al. [60], a complementary explanation could be a fire-induced cracking of long *n*-alkane chains, resulting in an increased abundance of short *n*-alkane chains. In fact, this can be clearly observed when comparing the *n*-alkane series of BAF and NAF (Figure 3a), where BAF is enriched in short and depleted in long *n*-alkane chains compared to NAF.

The BRA sample (0–5 cm depth) contained the highest relative abundance of Pol compounds (3.52 g/kg OC) compared to NAF and BAF (Table 2). This can be assigned to the long-term monoculture of *Brachiaria brizantha*, resulting in an enriched contribution of this vegetation to SOM chemistry [61]. This is supported by the remarkable depletion of Lip compounds (1.54 g/kg OC) in BRA compared to BAF and NAF (Table 2), attributed to the lower amount of epicuticular waxes in Brachiaria plants [62]. The maximum *n*-alkane C25 observed in BRA corroborates the dominance of the herbaceous vegetation to SOM, whereas in NAF and BAF the maximum chain length occurred in *n*-alkane C31 and C27, respectively (Figure 3a), which are more likely assigned to arboreal vegetation [49,63].

At 5–10 cm depth, the highest abundance of UACs was observed in NAF (4.46 g/kg OC, Table 2), which may point out to the presence of highly altered SOM. As previously mentioned, labile SOM can be easily decomposed in Amazon soils, thereby promoting a relative increment of less-labile compounds such as UACs, especially in subsurface soil layers. The lowest relative abundance of UACs was observed in BRA (1.86 g/kg OC, Table 2) and this may be explained by the following processes: (i) mechanical soil mobilization resulting in the incorporation of fresh SOM from upper layers into subsurface layers; (ii) the time lapse since forest-burning in the BRA site (23 years), allowing the depletion and leaching of fire-derived-UACs over time; and (iii) input of biomass with high C:N ratio (92:1) in BRA (approximately 4-fold higher than that of NAF and BAF), slowing down its decomposition (Table 1). The first hypothesis can be supported by the relatively high abundance of lignocellulose (Pol: 2.12 g/kg OC and Lig: 0.11 g/kg OC compounds) in BRA compared to BAF and NAF at 5–10 cm soil depth (Table 2). These findings agree with the occurrence of a mixture of OM chemical compounds from surface and deeper soil layers caused by soil mobilization for agricultural practices, resulting in a “dilution effect” of chemical compounds along the soil profile [36]. The second interpretation is supported by authors who reported leaching of defunctionalized pyrogenic C in profile of Brazilian Cerrado soils [64], and degradation of pyrogenic C in soils of the Brazilian Atlantic Forest [65] and Amazon [66].

Relatively high contribution of short chain *n*-alkanes was observed at the 5–10 cm soil depth in NAF, BAF and BRA (Figure 3b), indicating the existence of humidified SOM produced by microbial activity [49,67,68]. Interestingly, the existence of mid/long chain *n*-alkanes in the BRA sample, absent in NAF (Figure 3b and Appendix A), reinforces the lower molecular diversity of SOM within 0–10 cm soil depth in BRA.

Noticeably, the highest relative abundance of PAHs at 5–10 cm depth occurred in NAF (1.07 g/kg OC, Table 2). This is possibly associated with the transport by air of pyrogenic compounds from the adjacent BAF area followed by migration of these compounds to deeper soil layers. This process can be facilitated by the high mean annual rainfall (2280 mm) and the coarse texture of the soil in the studied area [69]. Alternatively, a selective preservation of recalcitrant C that remained stable for a long period of time compared with more labile organic compounds, may also explain the accumulation of PAH in NAF [69,70]. However, these interpretations require further studies for clarification.

In BAF (5–10 cm depth), the higher Lip abundance (2.44 g/kg OC) compared to BRA and NAF (Table 2) may result from the condensation of such compounds in deeper soil layers following an inverse temperature gradient due to the recent fire [68,71]. Accordingly, Doerr et al. [72] observed a significant increase in water repellency in the subsurface of fire-affected soils due to lipidic condensation on the surface of soil particles. In fact, a detailed analysis of *n*-alkane chains clearly revealed the high abundance of *n*-alkane chains between C21 and C29 in BAF at 5–10 cm depth, whereas in NAF, these compounds were absent (Figure 3b), corroborating this assumption.

The SOM at lower depths (40–50 cm) was predominantly composed of UACs, Lip, Pol and Pep (Table 2), and with a lower diversity of Py-GC/MS compounds (Appendix A). The remarkable and similar abundance of Pol compounds observed in NAF and BAF at 40–50 cm depth (Table 2) may have a microbial origin from extracellular polymeric substances containing abundant compounds derived from sugars, as furanose (Appendix A), as recently suggested by Miller et al. [45]. Possibly, the higher relative abundance of Pep compounds in BRA (1.44 g/kg OC) compared to NAF and BAF (0.84 and 0.38, respectively, g/kg OC) (Table 2), may also have a microbial origin due to the stimulation of microorganisms by the release of labile organic compounds by Brachiaria roots [73]. For instance, across the investigated soil layers, the relative abundance of Pep compounds in both NAF and BAF samples clearly decreased along with soil depth, whereas in BRA it barely changed from 0–5 to 40–50 cm depth (Table 2). Additionally, enhanced microbial activity may lead to the depletion of labile organic compounds, thereby increasing the relative abundance of less labile components.

The detailed analysis of *n*-alkane series for the 40–50 cm depth soil samples was not possible due to the existence of only 2 or 3 short *n*-alkane compounds in the BRA sample, while no alkyl compounds were found in NAF and BAF samples (Appendix A).

### 3.2. Shifts in SOM Composition Revealed by van Krevelen Subtraction Plots

Figure 4 depicts van Krevelen subtraction plots (BRA-NAF, BAF-NAF and BAF-BRA) of pyrolysis chemical families obtained for 0–5, 5–10 and 40–50 cm soil depth. These subtraction plots are aimed to allow easy visualization of chemical/molecular shifts in SOM when contrasting the evaluated sites and soil depths.

Subtraction plots of 0–5 cm depth soil samples revealed the accumulation of Lip- and condensed-like compounds in BAF in relation to NAF (BAF-NAF, Figure 4). This is in line with our previous findings revealing a stronger Aryl-C signal in the ^13^C NMR spectrum of BAF compared to NAF [15], most likely attributed to pyrogenic compounds [20,74]. Conversely, BRA promoted the accumulation of UACs and the depletion of Lip-like and hydroaromatic compounds in relation to NAF (BRA-NAF, Figure 4). The possible stimulation of microbial activity in BRA, e.g., via exudates from vigorous Brachiaria root systems may have stimulated the cyclization of alkyl compounds and the dehydrogenation of hydroaromatic molecules, as both processes may result in the formation of UACs. However, more conclusive studies are required.

The subtraction plot of BAF in relation to NAF for the 5–10 cm depth was similar to that of the 0–5 cm depth (BAF-NAF, Figure 4), i.e., accumulation of Lip- and condensed-like compounds. Overall, these plots reveal a typical shift in SOM molecular composition up to 10 cm depth in BAF, either exerted by the direct effect of fire, and/or by the transport of fire-derived pyrogenic compounds from 0–5 to 5–10 cm depth. Moreover, these plots highlight the strong contribution of pyrogenic compounds to the SOM of BAF, despite 20 months of forest regrowth since the slash-and-burn of the forest. High temperatures (>350 °C) close to the soil surface during fire in BAF [15] may have reduced soil microbial communities [24], and, coupled with the accumulation of compounds toxic to microorganisms such as PAHs in BAF after fire (Table 2), likely hindered microbial proliferation and recovery in BAF. Consequently, a slow incorporation of new biomass added to soil into SOM composition can be expected [24]. The recalcitrant nature of PAHs and their toxicity to soil microorganisms add explanation to the 19% higher TOC content in BAF compared to NAF at 0–5 cm depth (Table 1).

The subtraction plot of BRA in relation to NAF for the 5–10 cm depth was similar to the 0–5 cm depth, i.e., accumulation of Lig phenols and UACs and remarkable depletion of hydroaromatic molecules (BRA-NAF, Figure 4). Additionally, at 5–10 cm depth, a relatively small increase in alkyl compounds was detected in BRA in relation to NAF, whereas at 0–5 cm depth a depletion of these compounds occurred (BRA-NAF, Figure 4). This shift in SOM chemistry possibly resulted from the changing of vegetation and the mixing of soil layers due to agricultural practices for pasture cultivation [36].

Subtraction plots obtained for 40–50 cm soil samples differed considerably from those of upper depths. In BRA, accumulation of alkyl and hydroaromatic compounds was detected in relation to NAF (BRA-NAF, Figure 4), especially due to short chain moieties. These results indicate high microbial activity leading to a highly degraded SOM. In contrast, recent fire apparently induced de-hydration/condensation reactions, resulting in the accumulation of UACs in BAF in relation to NAF (BAF-NAF, Figure 4) [46]. The subtraction plot of BAF in relation to NAF (BAF-NAF, Figure 4) suggests that reduction reactions of alkyl compounds, i.e., the defunctionalization of polar Lip-like compounds, may have occurred in BAF [59].

## 4. Conclusions

A typical pattern of SOM composition affected by fire was observed in BAF (increased abundance of UACs and PAHs and depletion of Pol compounds), especially at 0–5 cm depth. Additionally, BAF samples were found to be enriched in Lip compounds, possibly due to the deposition on soil of partially charred biomass immediately after fire or later due to post-fire plant stress/mortality. Our results suggest that fire may have led to the condensation of Lip compounds at 5–10 cm following an inverse temperature gradient, as evidenced by an increased abundance of Lip and the presence of long *n*-alkane chains at 5–10 cm in BAF.

Despite 20 months of post-fire forest regrowth and the accumulation of fresh litter on soil in BAF, this had a less pronounced contribution to SOM composition. Instead, the lingering effects of fire on SOM composition were dominant, indicating that the SOM composition in BAF has minimally recovered. This could be associated with the accumulation of compounds toxic to microorganisms (such as PAHs) in BAF, hindering the proliferation of beneficial microbial communities and the incorporation of newly added forest biomass to soil into SOM composition.

The *n*-alkane series of BRA samples revealed a maximum homologous of C25, which can be attributed to the dominant contribution of the herbaceous vegetation (Brachiaria) to SOM. The pronounced contribution of Brachiaria vegetation to SOM molecular composition was also confirmed by the low abundance of Lip in BRA, probably resulting from the lower amount of epicuticular waxes in this vegetation. Additionally, the depletion of UACs and PAHs in BRA compared to BAF, especially at 0–5 cm, strongly suggests that fire-derived-pyrogenic C was lost by decomposition and/or leaching and may be less relevant in contributing to SOM chemistry in BRA. The loss of forest-derived SOM in BRA as shown in this study, together with the lower addition of biomass to soil in BRA compared to the forest sites, may partially explain the lower TOC content observed in BRA compared to NAF and BAF.

To our knowledge, this is the first work where Py-GC/MS-derived 3D van Krevelen diagrams were used to reveal shifts in SOM molecular composition in AF soils caused by fire and post-fire forest regrowth or pasture. Overall, our results are in line and added chemical detail to our previous findings using ^13^C NMR. Additionally, this analytical–graphical approach proposes explanations and hypotheses for the considerable high contributions of UACs and PAHs to SOM in BAF (i.e., direct effects of fire) and in NAF where this was not expected (possibly inherited from fire-event, or probably aerial transportation–deposition of pyrogenic compounds from the adjacent burned site, BAF).

We conclude that, despite the fast regrowth of AF after fire, the SOM underwent a minimal recovery, and several years may be required to regain soil resilience and recover natural protection of AF against wildfire. Our findings also draw attention to shifts in SOM chemistry and eventual toxicity to microorganisms in unburned sites caused by fire in adjacent areas, likely due to the transportation of pyrogenic compounds by air to the native forest floor. This topic has not been acknowledged with due importance and requires urgent studies.

## Figures and Tables

**Figure 1 ijerph-20-03485-f001:**
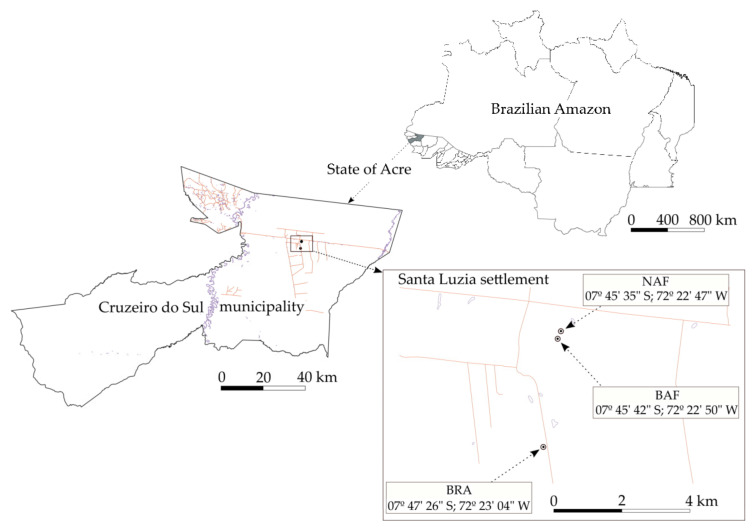
Location of the evaluated sites: native Amazon Forest (NAF), burned Amazon Forest (BAF) and Brachiaria pasture (BRA) in Cruzeiro do Sul city, Acre State, Brazil. Adapted from Leal et al. [15] based on a CC_BY license adoption.

**Figure 2 ijerph-20-03485-f002:**
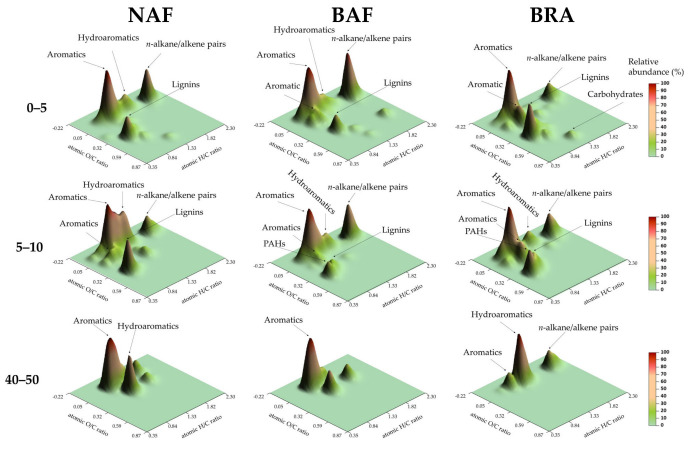
Surface density maps displaying cumulative abundances of Py-GC/MS products plotted in the space defined by the atomic H/C (*x*) and O/C (*y*) ratios in a van Krevelen diagram and the chromatographic yields as total abundances (*z*). Labels on the plots indicate major groups of organic compounds. Soil samples were collected at 0–5, 5–10 and 40–50 cm depth at the native Amazon Forest (NAF), burned Amazon Forest (BAF) and Brachiaria (BRA) site.

**Figure 3 ijerph-20-03485-f003:**
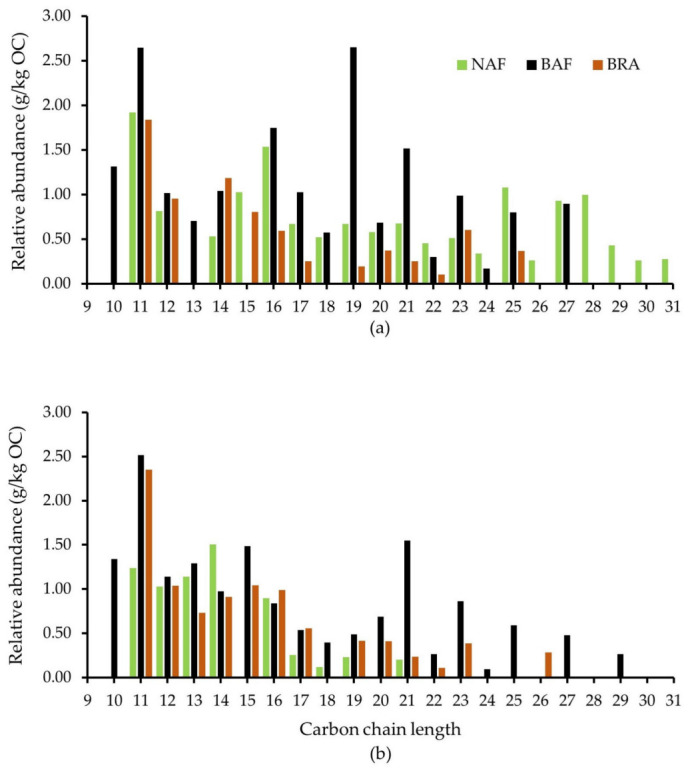
Abundance of *n*-alkanes chains relativized to the organic carbon content (g/kg OC) of soil samples collected at the native Amazon Forest (NAF), burned Amazon Forest (BAF) and Brachiaria (BRA) site at 0–5 cm (**a**) and 5–10 cm depth (**b**).

**Figure 4 ijerph-20-03485-f004:**
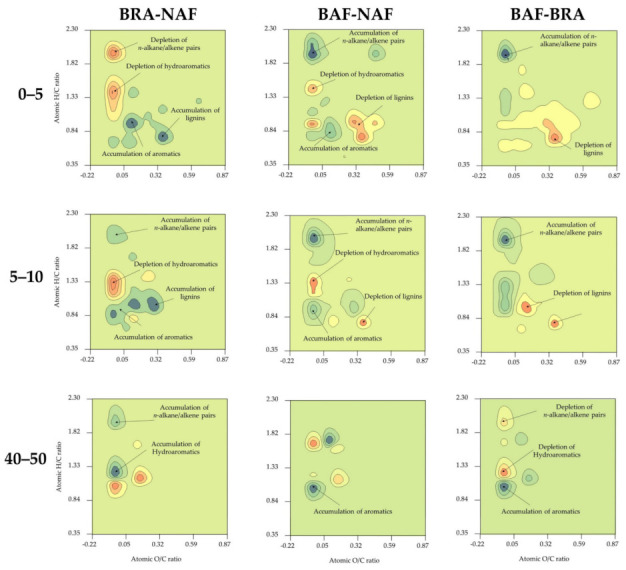
Subtraction density maps illustrating differences in abundance of pyrolysis products between soil samples collected at the native Amazon Forest (NAF), burned Amazon Forest (BAF) and Brachiaria (BRA) site at 0–5, 5–10 and 40–50 cm depth. Maps are represented in the space defined by the H/C and O/C atomic ratios of the corresponding compounds. Enrichment of compounds is shown in blue anddepletion in red.

**Table 1 ijerph-20-03485-t001:** Values of pH, available phosphorous (P) and potassium (K^+^), exchangeable calcium (Ca^2+^), magnesium (Mg^2+^) and aluminum (Al^3+^) content, potential acidity (Al+H), effective cation exchange capacity (CECe) and total organic carbon content (TOC) of soil samples collected at 0–5, 5–10 and 40–50 cm depth at the native Amazon Forest (NAF), burned Amazon Forest (BAF) and Brachiaria (BRA) site.

Site	pH	P	K^+^	Ca^2+^	Mg^2+^	Al^3+^	Al+H	CECe	TOC *
		(mg dm^−3^)		(cmol_c_ dm^−3^)					g kg^−1^
	0–5 cm								
NAF	3.8	4.1	26.0	0.1	0.1	2.2	15.4	2.5	14.5
BAF	4.7	3.1	53.0	1.5	0.5	1.3	7.7	3.4	17.9
BRA	4.7	2.1	47.0	0.8	0.4	0.5	3.5	1.8	10.1
	5–10 cm								
NAF	4.0	2.1	17.0	0.1	0.1	2.5	15.4	2.7	10.4
BAF	4.3	3.1	32.0	0.5	0.2	2.6	9.7	3.4	10.3
BRA	4.9	1.6	25.0	1.0	0.2	0.8	3.5	2.1	8.0
	40–50 cm								
NAF	4.6	1.1	11.0	0.1	0.1	4.5	17.3	4.7	3.4
BAF	4.4	0.8	39.0	0.1	0.1	3.7	12.3	4.0	4.6
BRA	4.5	0.4	22.0	0.1	0.1	3.4	7.7	3.7	2.9

* Adapted from Leal et al. [15] based on a CC_BY license adoption.

**Table 2 ijerph-20-03485-t002:** Abundance of chemical families released by Py-GC/MS of soil samples collected at 0–5, 5–10 and 40–50 cm depth at the native Amazon Forest (NAF), burned Amazon Forest (BAF) and Brachiaria pasture (BRA) site. Values are relativized to the organic carbon content of the sample (g/kg OC).

Chemical Family *	NAF	BAF	BRA	NAF	BAF	BRA	NAF	BAF	BRA
	g/kg OC								
	0–5 cm			5–10 cm			40–50 cm		
Lip	4.10	5.26	1.54	1.59	2.44	1.44	0.30	0.00	0.62
UACs	3.21	6.03	2.26	4.46	3.66	1.86	1.47	3.45	0.70
Lig	0.65	0.37	0.31	0.00	0.00	0.11	0.00	0.00	0.00
N-comp	0.09	0.28	0.29	0.63	0.07	0.16	0.00	0.00	0.00
PAHs	1.16	1.57	0.42	1.07	0.93	0.74	0.00	0.00	0.04
Pep	2.15	2.45	1.74	0.63	1.63	1.56	0.84	0.38	1.54
Pol	3.14	1.95	3.52	2.03	1.57	2.12	0.79	0.77	0.00

* Lip = lipids, UACs = unspecific aromatic compounds, Lig = lignin-derived, N-comp = nitrogen compounds, PAHs = polycyclic aromatic hydrocarbons, Pep = peptides, Pol = polysaccharide-derived.

## Data Availability

The data presented in this study are available in the article and Appendix A.

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
