# Peer review of "Soil Organic Matter Molecular Composition Shifts Driven by Forest Regrowth or Pasture after Slash-and-Burn of Amazon Forest"

_ijerph, 2023, doi:10.3390/ijerph20043485_

Round 1

Reviewer 1 Report

Comments to “Soil Organic Matter Molecular Composition Shifts Driven by Forest Burning Followed by Forest Regrowth or Pasture in Northern Brazilian Amazon”

The authors investigated changes in SOM composition at the molecular level after the fire in forest and grassland ecosystems in Northern Brazilian Amazon. They found that the remaining effects of fire significantly affect SOM composition compared to a native Amazon Forest. These results are exciting and need to be published. However, the results are too simple. The author did not draw enough conclusions despite doing many analyzes. Writing is another major concern. So, I think the MS should be published after major revisions.

Title and Abstract

1.1   Title: It’s not easy for potential readers to immediately understand the author’s meanings by the title. Maybe it’s better to rephrase the title related to “Forest Burning Followed by Forest Regrowth or Pasture.” The title should be changed, making it shorter and more transparent.

1.2   Abstract: Too many abbreviations and some important results were unambiguous. The main conclusions and key implications need to be clarified.

1.3   Keywords: Too many keywords. The keywords need more consideration.

Introduction section:

2.1   There is some scattered information throughout the text. The authors should increase and organize the information, highlighting the necessity of the research and the main contribution of the former studies.

2.2   Only closely related articles should be displayed here.

Materials and methods section

3.1 You mentioned 0-5, 5-10, and 40-50 cm soil layers; why did you choose these three layers?

3.2 More details are needed about the sampling sites, especially human interventions.

Results and Discussion:

4.1 In general, the authors should make an effort to reduce these sections.

Round 2

Reviewer 1 Report

The authors have deepened the treated topic by improving the discussion of the results and highlighting some applicative aspects in the conclusions. So I think the MS can be published in the journal.